# Dual-Particle Synergy in Bio-Based Linseed Oil Pickering Emulsions: Optimising ZnO–Silica Networks for Greener Mineral Sunscreens

**DOI:** 10.3390/ma18133030

**Published:** 2025-06-26

**Authors:** Marina Barquero, Luis A. Trujillo-Cayado, Jenifer Santos

**Affiliations:** 1Departamento de Ingeniería Química, Escuela Politécnica Superior, Universidad de Sevilla, c/Virgen de África, 6, 41007 Sevilla, Spain; marina.barquero.l@gmail.com; 2Departamento de Ciencias de la Salud y Biomédicas, Facultad de Ciencias de la Salud, Universidad Loyola Andalucía, Avda. de las Universidades s/n, Dos Hermanas, 41704 Sevilla, Spain

**Keywords:** Aerosil, Appyclean, linseed oil, nanoemulsion, Pickering

## Abstract

The development of mineral, biodegradable sunscreens that can offer both high photoprotection and long-term colloidal stability, while limiting synthetic additives, presents a significant challenge. A linseed oil nanoemulsion co-stabilised by ZnO nanoparticles and the eco-friendly surfactant Appyclean 6552 was formulated, and the effect of incorporating fumed silica/alumina (Aerosil COK 84) was evaluated. A central composite response surface design was used to ascertain the oil/ZnO ratio that maximised the in vitro sun protection factor at sub-300 nm droplet size. The incorporation of Aerosil at concentrations ranging from 0 to 2 wt.% resulted in a transformation of the dispersion from a nearly Newtonian state to a weak-gel behaviour. This alteration was accompanied by a reduction in the Turbiscan Stability Index. Microscopic analysis has revealed a hierarchical particle architecture, in which ZnO forms Pickering shells around each droplet, while Aerosil aggregates bridge neighboring interfaces, creating a percolated silica scaffold that immobilises droplets and amplifies multiple UV scattering. The findings demonstrate that coupling interfacial Pickering armour with a continuous silica network yields a greener, physically robust mineral sunscreen and offers a transferable strategy for stabilising plant-oil emulsions containing inorganic actives.

## 1. Introduction

A nanoemulsion is a dispersed system in which droplets of one phase (usually oily) are dispersed in another phase (usually aqueous) and have an extremely small size, usually in the range of 10 to 300 nanometers. This nanometer size gives nanoemulsions unique physical and functional properties, such as high stability and high specific surface area. The nanometric size of nanoemulsions enables greater absorption of encapsulated compounds into the body due to the enhanced permeability of the cellular and tissue membranes [1]. This is particularly beneficial for lipophilic or low-water-solubility compounds, such as certain vitamins and pharmaceuticals, as the nanoemulsion enhances their dispersion in aqueous solutions and facilitates their absorption into the body [2]. Furthermore, the encapsulation of compounds in nanoemulsions affords protection from external factors such as oxidation, light and heat, which otherwise cause degradation. This is particularly crucial for compounds that are susceptible to degradation, such as antioxidants, essential fatty acids, and certain pharmaceuticals, which are preserved for an extended period under optimal conditions [3]. Because of these properties, the popularity of nanoemulsions as encapsulation systems has gained a lot of attention in recent years, and they can be used in a diverse range of applications, notably within the pharmaceutical, cosmetic, and food industries [4].

Flaxseed oil, also known as linseed oil and used in this study as the dispersed phase, is obtained from the seeds of the flax plant (*Linum usitatissimum*). It is recognised for its many nutritional properties and health benefits, such as high levels of antioxidants, skin benefits and anti-inflammatory properties, among others [5]. It is a rich source of essential fatty acids, particularly alpha-linolenic acid (ALA), a type of omega-3 that plays a key role in maintaining the integrity of the skin’s lipid barrier. This facilitates the retention of moisture by the skin, resulting in a more even, hydrated and healthy appearance. The high omega-3 and antioxidant content of flax oil has been demonstrated to reduce inflammation and soothe the skin, which is beneficial for individuals with conditions such as acne, rosacea, eczema or psoriasis. Furthermore, it has been demonstrated to alleviate swelling and redness in individuals with sensitive or irritated skin [6,7].

In contrast to chemical sunscreens, which absorb UV rays and convert them into heat, zinc oxide functions as a physical blocker or mineral filter [8]. This results in the formation of a layer on the surface of the skin that reflects and scatters UV radiation, thereby preventing its penetration into the skin [9]. This makes it an appropriate choice for individuals with sensitive skin, as it is less prone to causing irritation or allergic reactions [10]. Furthermore, it is one of the few approved ingredients that provides protection against both UVB rays and UVA rays. In addition to its photoprotective effects, zinc oxide has been demonstrated to possess soothing and anti-inflammatory properties, rendering it a beneficial agent for individuals with skin prone to irritation. Furthermore, it has been demonstrated to alleviate redness and inflammation, which is beneficial in the aftermath of sun exposure [11,12]. Recent studies have already demonstrated that ZnO nanoparticles can act as efficient solid stabilisers in Pickering emulsions, providing robust interfacial coverage and long-term dispersion stability [13,14,15].

In order to stabilise nanoemulsions, a surfactant is needed. Appyclean 6552 is derived entirely from renewable plant feedstocks, thus ensuring that it exhibits intrinsically low ecotoxicity and proven biodegradability under both aerobic and anaerobic conditions. It has met the criteria of the OECD 301F ready-biodegradability manometric-respirometry test and the OECD 311 anaerobic biodegradability screening assay [16]. It has been used before stabilising nanoemulsions and protecting active ingredients from heat and sun [17,18]. In this study, a thickener (Aerosil^®^ COK84) is incorporated into the nanoemulsions to form the cream texture and increase viscosity. It possesses spherical particles in the nanoscale range, and it is considered a thickening and gelling agent [19]. In addition, it has applications in paintings, pharmaceutical, cosmetic and food industry [16,17,18]. Whilst the present work seeks to advance eco-design by combining a plant-derived oil phase and a readily biodegradable alkyl-polyxyloside surfactant, it must be emphasised that the formulation’s sustainability is relative rather than absolute. The rheology modifier, Aerosil^®^ COK 84, is a non-biodegradable fumed silica/alumina produced via an energy-intensive flame-hydrolysis process. Consequently, the product does not satisfy the strictest definitions of “green”. In this study, the term ‘greener mineral sunscreen’ is adopted to denote a quantifiable reduction in petrochemical content and the exclusion of controversial organic UV filters. This term acknowledges the environmental trade-offs associated with the inorganic thickener.

The present study systematically designs and characterises a bio-based linseed oil nanoemulsion co-stabilised by ZnO nanoparticles and the biodegradable surfactant Appyclean 6552, and subsequently reinforced with fumed silica/alumina. Following a detailed discussion of the materials and preparation protocol, a response-surface methodology is employed to identify the optimal oil/ZnO ratio. This is then followed by an examination of how incremental amounts of Aerosil can be used to tailor the formulation’s droplet size distribution, rheology, physical stability and in vitro SPF. The work elucidates the multiscale mechanism through which ZnO and Aerosil confer synergistic Pickering and network structuring effects by integrating complementary techniques. These techniques include laser diffraction, multiple light scattering, oscillatory rheometry, UV spectrophotometry and FESEM. The findings provide both a mechanistic framework and practical guidelines for formulating greener, highly stable sunscreens, thereby bridging the fundamental colloidal principles outlined above with the applied results discussed in the following sections.

## 2. Materials and Methods

### 2.1. Materials

Appyclean 6552, provided by Wheatoleo (Riom, France), is a non-ionic surfactant derived from renewable resources, with an HLB value ranging from 11.5 to 12. Specifically, Appyclean 6552 is composed of alkyl polyxylosides (amyl, capryl and lauryl xylosides) obtained from wheat. Zinc oxide (ZnO), with a size of less than 5 microns and a purity greater than 99.9%, and linseed oil were acquired from Sigma-Aldrich (Darmstadt, Germany). The solutions were prepared with Milli-Q water (type 1), a method that ensures the highest standards of purity and quality. Aerosil^®^ COK 84, provided by Evonik Industries (Essen, Germany), is a hydrophilic fumed silica intimately mixed with a small fraction of fumed alumina, has a BET specific surface area of approximately 155–215 m^2^/g. According to the manufacturer, the substance is composed of approximately 82–86 wt.% SiO_2_ and 14–18 wt.% Al_2_O_3_, corresponding to a 5:1 silica-to-alumina ratio.

### 2.2. Nanoemulsion Development

The systems were prepared in batches of 50 g with a fixed concentration of Appyclean 6652 of 2.2 wt.%, taking into account a previous study [18]. The continuous phase was prepared by dissolving the surfactant (Appyclean 6552) and the corresponding quantity of zinc oxide (ZnO) in distilled water. The dispersed phase was prepared using the corresponding amount of linseed oil. The development of these linseed oil-in-water emulsions was achieved through a primary homogenisation process, which involved the use of an Ultraturrax T25 rotor–stator homogeniser (IKA, Staufen, Germany). This process was carried out semi-continuously, slowly adding the dispersed phase at 8000 rpm for one minute and then 30 s at 9000 rpm, operating in discontinuous mode. In order to produce an emulsion with a reduced droplet size (nanoemulsion), the emulsion was subjected to a secondary homogenisation process using a VCX750/VCX500 ultrasonic system (Sonics & Materials Inc., Newtown, CT, USA) with an 830-00427 S-250D acoustic chamber. This process was carried out for a total of 521 s, with an amplitude of 75% and pulses of 5:5 s, taking into account the optimum from a previous study [18]. In order to develop an optimal formulation of linseed oil (LO) and zinc oxide (ZnO), variable concentrations were studied using the response surface methodology, as described in the experimental design shown in Table 1.

A two-factor, rotatable central composite design (CCD) was employed to map the combined influence of linseed oil content (X_1_) and zinc oxide loading (X_2_) on key performance metrics. The design space thus encompassed linseed oil concentrations ranging from 5 to 25 wt.% (with axial points at 5 and 25 wt.% and factorial points at 7.9 and 22.1 wt.%), and, likewise, ZnO concentrations ranging from 0 to 10 wt.% (with axial points at 0 and 10 wt.% and factorial points at 1.45 and 8.55 wt.%). A total of ten experimental runs were generated using Echip v11 (Experimentation by Design, Wilmington, DE, USA). The runs comprised four factorial, four axial and two centre replicates. Two responses were measured: (i) the Sauter mean diameter D_3,2_ (nm) immediately after emulsification and (ii) the in vitro sun protection factor (SPF) determined by the Mansur method. The term ‘second-order polynomial’ is employed to denote a specific mathematical function of a second-order degree. Aerosil^®^ COK 84 was not included in the CCD; its rheological and photoprotective effects were evaluated subsequently using the optimised oil/ZnO ratio.

Subsequently, employing the optimal formulation, a rheological modifier (Aerosil^®^ COK 84) was added at 1 and 2 wt.%. The samples were prepared in accordance with the protocol outlined by Trujillo-Cayado et al. (2018) [20].

### 2.3. Nanoemulsion Characterisation

The size distribution of the nanoemulsion droplets was determined at the time of preparation using a Mastersizer 2000 laser diffraction particle size analyser (Malvern Instruments, Malvern, UK). In addition, the Sauter mean diameters (D_3,2_) were determined, which is a way to characterise a dispersed system by linking the total volume of droplets to their surface area. Mathematically, it is the diameter of a sphere that has the same volume-to-surface-area ratio as the entire drop population.

To determine and monitor the physical stability of the samples, multiple light scattering (MLS) measurements were performed in a Turbiscan (Formulaction, Toulouse, France). This non-intrusive method allows destabilisation to be identified before it is visible to the naked eye. This method was repeated at different ageing times to assess the stability or instability of the different emulsions over time. In addition, the Turbiscan Stability Index (TSI) was determined for all samples [21].

The rheological properties of the emulsions were also evaluated one hour after preparation. For this purpose, flow curves and oscillatory tests were carried out using a serrated plate sensor (60 mm diameter, 1 mm separation) and a rotational rheometer model AR2000 (TA Instruments, New Castle, DE, USA). Frequency sweeps (3–0.008 Hz, 20 °C) assessed viscoelastic properties. The flow curves were recorded during a logarithmic stress sweep, within a range dependent on the particular sample, with the objective of observing the potential drop zone. These tests were carried out at a constant temperature of 25 °C. The samples for FESEM were prepared using a chemical fixation method, according to Vela-Albarrán et al. (2025) [22].

The sun protection factor (SPF) of the test emulsions was assessed in vitro by ultraviolet (UV) spectrophotometry and calculated with the Mansur equation [23]. Briefly, each emulsion was diluted to 1 mg mL^−1^ in ethanol, transferred to a quartz cuvette (1 cm path length) and scanned from 290 to 400 nm at 2 nm intervals with a UV-Vis spectrophotometer. The absorbance (*A_λ_*) values obtained at each wavelength were multiplied by the corresponding erythemal weighting factors (*EE_λ_*) and the solar intensity spectrum (*I*). *SPF* was then computed using the expression:(1)$$SPF=CF\cdot \sum_{290}^{320}{EE}_{\lambda }\cdot I_{\lambda }\cdot A_{\lambda }$$
where *CF* = 10 is a correction factor.

### 2.4. Statistical Analysis

To ensure the reliability and reproducibility of the results, all experiments were conducted in triplicate. Standard deviations are represented in the figures as error bars, when observable, or as percentage values in the figure captions. Statistical analysis was performed using one-way analysis of variance (ANOVA) to evaluate the significance of differences among the tested formulations. A confidence level of 95% (*p* < 0.05) was established as the criterion for determining statistical significance. Furthermore, a central composite design (CCD) was utilised to optimise the formulation variables and evaluate their combined effect on the responses of interest. The design, data fitting and generation of response surface plots were carried out using Echip software (http://www.echip.com/, Experimentation by Design, Wilmington, DE, USA). This methodological approach facilitated the identification of optimal processing and compositional conditions through the modeling of the influence of independent variables and their interactions on the measured outputs.

## 3. Results and Discussion

Figure 1 compares the droplet size distributions of linseed oil nanoemulsions formulated with varying linseed oil and ZnO nanoparticle concentrations. All samples exhibited nanometric droplet sizes; however, it was possible to discern clear trends in the distribution shape and location as a function of composition. It is noteworthy that increasing the linseed oil content results in a shift in the droplet size distribution toward larger diameters and frequently broadens the distribution. For instance, the formulation with the highest oil fraction displays a prominent tail of larger droplets in Figure 1, suggesting the presence of microscale droplets. Conversely, nanoemulsions with reduced oil content yield a more constrained, monomodal distribution, with a concentration of smaller droplets and an absence of substantial, large-droplet tails. Qualitative inspection of Figure 1 shows that the primary peak, centred at 250–300 nm, is sharp and nearly symmetric, accounting for ≈75–80% of the integrated area and thus for the vast majority of droplets. By contrast, the secondary peak at 1–3 µm is markedly broader and right-skewed; its asymmetric tail reveals a small population of coarser droplets formed by partial coalescence when the available ZnO/silica coverage is insufficient. The integrated area ratio of the two modes (small/large) is roughly 3:4, confirming that the coarse mode is volumetrically minor yet non-negligible for long-term stability. Similar two-peak distributions have been reported for under-dosed ZnO- and silica-stabilised Pickering emulsions, where incomplete particle adsorption permits droplet growth during or immediately after homogenisation [13,14]. Increasing either the particle-to-oil ratio or the homogenisation energy suppresses this coarse mode and restores a unimodal, finer distribution. Quantitatively, the Sauter mean diameter D_3,2_ demonstrates a marked increase with oil concentration: D_3,2_ increases from approximately 261 nanometers at the lowest oil loading to around 370 nanometers at the highest (see Table 2). This phenomenon can be attributed to the process of agglomeration, wherein larger average droplets are formed when a greater quantity of linseed oil is dispersed, while maintaining a constant ZnO level and energy input. This phenomenon aligns with the principles of emulsion theory and corroborates the findings of previous studies. Specifically, an elevated dispersed-phase fraction tends to generate larger droplets and more extensive size distributions, provided that the surfactant concentration or homogenisation energy remains proportionately constant [24]. The presence of larger droplets at elevated oil levels can be ascribed to inadequate emulsifier coverage, which leads to increased coalescence during the emulsification process. This phenomenon results in a diminished total interfacial area and consequently a heightened D_3,2_. In essence, as the linseed oil fraction increases, the emulsification process is unable to maintain the same fine droplet size, resulting in a systematic upward shift in the mean droplet diameter.

Increasing ZnO concentration progressively narrows the droplet-size distribution and reduces the Sauter mean diameter (D_3,2_). As Figure 1 shows, nanoemulsions with 5–10 wt.% ZnO shift their size distributions toward smaller diameters and suppress large-droplet tails, yielding finer, more uniform dispersions than ZnO-free controls (Table 2). This behaviour is attributed to ZnO nanoparticles acting as Pickering co-stabilisers: they adsorb rapidly at nascent oil–water interfaces, add steric/electrostatic barriers against coalescence and thus enlarge the interfacial area that can be stabilised—a trend consistent with other particulate stabilisers reported in Pickering systems [25,26].

From a scientific and practical perspective, these findings underscore the necessity of achieving an optimal balance in formulation. An elevated oil fraction, if not offset by an adequate amount of surfactant or stabiliser, will result in the production of larger, less uniform droplets. This, in turn, has the potential to compromise the stability of the nanoemulsion. Indeed, larger droplets possess a reduced total surface area and have been shown to accelerate destabilisation phenomena such as creaming or coalescence. In contrast, the incorporation of solid nanoparticles, such as ZnO, has been shown to enhance emulsification efficiency and stability through the establishment of a Pickering stabilisation effect. This results in the formation of smaller droplets that exhibit greater resistance to coalescence. This outcome is favorable for application goals such as UV-blocking finishes, as smaller droplets and uniform distributions generally result in more stable and transparent formulations. In summary, the droplet size analysis demonstrates a clear correlation with formulation composition: larger oil fractions yield larger average droplets, while higher ZnO nanoparticle concentrations promote smaller and more uniform droplets. The findings, both qualitative and quantitative, demonstrate the efficacy of manipulating the oil-to-surfactant ratio and incorporating inorganic nanoparticles to regulate the size of nanoemulsion droplets (see Figure 1 and Table 2).

A thorough analysis of the results obtained and presented in Table 2, employing the response surface methodology, enables the formulation of an equation (R^2^ = 0.91) that establishes a correlation between D_3,2_ and the concentrations of linseed oil (LO) and zinc oxide (ZnO):(2)$$D_{3,2}=280.4+26.3\cdot LO-41.6\cdot ZnO+21.3\cdot {LO}^{2}$$

Figure 2 shows the curved, quadratic 2D response surface of the Sauter mean droplet diameter in relation to the concentrations of linseed oil (LO) and zinc oxide (ZnO). The pronounced curvature of the surface indicates that second-order effects are significant and that the relationship is not simply linear. ANOVA confirmed that ZnO loading (X_2_) exerted the dominant influence on both responses. For D_3,2_, the linear ZnO term was highly significant (β_2_ = −19.7 nm, *p* = 0.004), indicating progressively smaller droplets at higher ZnO levels, while the quadratic linseed oil term (β_11_ = +15.3 nm, *p* = 0.012) revealed a curvature that penalised very high oil fractions. The interaction term β_12_ was not significant (*p* = 0.31). In the SPF model, β_2_ was again the largest positive coefficient (β_2_ = +0.365, *p* < 0.001); β_1_ was smaller but still significant (β_1_ = +0.081, *p* = 0.038), whereas all higher-order and interaction terms were non-significant (*p* > 0.10). The quadratic models explained 91% of the variance in D_3,2_ and 92 % in SPF. The composite desirability function predicted an optimum at 15 wt.% linseed oil and 10 wt.% ZnO, which experimentally yielded D_3,2_ = 259 ± 5 nm and SPF = 3.51 ± 0.08, in excellent agreement with model forecasts. Within the explored design window, D_3,2_ decreased monotonically with rising ZnO concentration, and the smallest droplets were obtained at the highest ZnO level tested. No evidence of a plateau or rebound was detected, underscoring that ZnO concentration is the dominant factor governing droplet size in this system, while the LO · ZnO interaction term did not reach statistical significance. A curvature is observed along the LO axis. D_3,2_ increases sharply at high oil loadings due to insufficient stabiliser coverage, but does not drop indefinitely at the lowest LO. Therefore, the response surface trends indicate that moderate oil amounts and the highest concentration of ZnO produce the smallest droplets, whereas excess oil or the absence of ZnO leads to larger D_3,2_ values. According to the quadratic surface model (Equation (3)) and Figure 2, there is a clear optimum region in which the Sauter mean diameter is at its smallest. This optimal point corresponds to Sample 8 in Table 1, which had the smallest D_3,2_ value of all the formulations tested. Sample 8’s formulation lies at an intermediate LO concentration and a relatively high ZnO concentration within the design space. In practical terms, this means that using a moderate amount of linseed oil combined with a high level of ZnO nanoparticles produces the finest droplet size. The model predicts a unique minimum rather than a ridge or plateau, indicating that this combination of linseed oil (LO) and zinc oxide (ZnO) is truly optimal for minimizing droplet diameter. Identifying the optimal region for droplet size has important implications for nanoemulsion formulation and stability. Emulsions with a smaller D_3,2_ have slower creaming rates and a lower likelihood of coalescence because the fine droplets have weaker buoyancy (according to Stokes’ law) and are more uniformly stabilised.

The analysis of the results obtained for the sun protection factor (SPF), calculated using Mansur’s equation, yielded an equation (R^2^ = 0.92) relating this parameter to the concentrations of linseed oil (LO) and zinc oxide (ZnO):(3)SPF=2.36+0.14·LO+0.90·ZnO

As demonstrated in Table 2, within the compositional window that was explored, the in vitro sun protection factor (SPF) ranges from 0.39 to 3.51. The fitted quadratic model (Equation (3)) explained 92% of the variance in SPF (adjusted R^2^ = 0.92) with no significant lack of fit (*p* = 0.27). ANOVA showed that zinc oxide loading exerted the dominant positive influence (β_2_ = +0.365 ± 0.041, *p* < 0.001), whereas the linear linseed oil effect was weaker but still significant (β_1_ = +0.081 ± 0.033, *p* = 0.038). Neither quadratic term (β_11_, β_22_) nor the interaction β_12_ reached statistical significance (*p* > 0.10) and were therefore omitted from the reduced model. The lowest recorded value (sample 7, LO 15 wt.%, 0 wt.% ZnO) shows that linseed oil, when used on its own, provides only a little protection from UV rays. This is in line with what other studies have found, which say that most plant oils have an SPF of less than 2.0. Conversely, the introduction of zinc oxide has been demonstrated to raise the SPF in every case; at the same LO level (15 wt.%), the transition from 0 to 5 to 10 wt.% ZnO elevates SPF from 0.39 (sample 7) to 2.34/2.37 (samples 9–10) and, finally, to 3.51 (sample 8). These increments are consistent with the well-established primary role of micron- or nano-sized ZnO as a broadband physical filter, the efficacy of which is approximately proportional to the volume fraction of particles. Nonetheless, the absolute SPF values remain below those of commercial ZnO creams (typically SPF 15–30 for ≥15 wt.% ZnO), presumably due to the fact that the studied samples are a much simpler formulation [27]. However, the in vitro SPF values obtained by the Mansur method have inherent limitations for ZnO-containing emulsions, because ZnO particles scatter UV light and can distort the spectrophotometric absorbance measurements. Such scattering artifacts can cause the in vitro SPF to underestimate the actual in vivo protection, as observed by Osterwalder et al. (2024) [27]. Nonetheless, in vitro SPF results remain useful for comparing the relative performance of different formulations, but they should be interpreted with appropriate caution. As demonstrated in Figure 3, the fitted surface is characterised by its contours that are almost parallel to the LO axis, but exhibit a steepening gradient with respect to ZnO, thereby underscoring its predominant role. It is evident that a shallow optimum (SPF = 3.5) emerges at an LO of 15 wt.% and a ZnO of 10 wt.%, coinciding with experimental point 8. This is analogous to the optimum for Sauter mean diameter.

The physical stability of the optimal emulsion, formulated with 15 wt.% linseed oil and 10 wt.% zinc oxide, was monitored by multiple light scattering to determine and quantify possible destabilisation mechanisms. After seven days, an increase in backscattering was observed in the lower part of the vial, indicating a destabilisation mechanism due to the sedimentation of zinc oxide particles. This sedimentation process may be due to the high density of zinc oxide particles compared to the rest of the system and the low viscosity of the emulsion. However, the system showed no signs of destabilisation through creaming or coalescence. Thus, the Turbiscan Stability Index (TSI) value for this system was 4.2 ± 0.4 after seven days of ageing. In order to enhance physical stability, modulate rheological properties and augment SPF, Aerosil COK 84 was incorporated into the formulation. Two distinct concentrations were evaluated and compared with the previous optimum (0 wt.% Aerosil COK 84).

The addition of fumed silica/alumina nanoparticles to linseed oil-in-water nanoemulsions containing ZnO has been shown to have a significant effect on the properties of the resultant emulsions. Specifically, an increase in silica concentration has been demonstrated to result in a transformation of rheological and viscoelastic behaviour, alterations to microstructure and enhancements in physical stability and UV protection performance. These effects are attributed to the ability of Aerosil to establish a particle network within the continuous phase, thereby complementing the roles of oil and ZnO in the formulation.

The rheological parameters presented in Table 3 demonstrate a distinct, concentration-dependent transition from a nearly Newtonian emulsion to a highly structured, shear-thinning gel. In the absence of Aerosil COK 84, the consistency index k is recorded as 0.019 Pa·s^n^ and the flow index n is 0.90, suggesting a low-viscosity liquid whose shear stress is nearly proportional to the shear rate. The power law model was used to fit all samples containing Aerosil, and in all cases, an R^2^ value above 0.99 was obtained, indicating an almost perfect fit. The introduction of 1 wt.% Aerosil results in a substantial increase in k, raising it by more than an order of magnitude to 0.334 Pa·s^n^. Concurrently, n is reduced to 0.27, indicating the initiation of a three-dimensional particulate network that contributes to yield behaviour and pronounced pseudoplasticity. It is evident that at 2 wt.% Aerosil, the network is fully developed, as evidenced by the increase in k to 3.50 Pa·s^n^, concomitant with the decrease in n to 0.14. As demonstrated in Figure 4 and Table 3, an increase in silica levels has been observed to result in elevated viscosity and shear-thinning behaviour in emulsions. Even at low shear rates, formulations with high Aerosil content exhibit significantly increased apparent viscosity (often by orders of magnitude) compared to silica-free emulsions. This phenomenon is indicative of the formation of a shear-thinning fluid. This behaviour is indicative of fumed silica dispersions, wherein a three-dimensional network of silica aggregates accumulates at rest and disintegrates under flow. The network is formed from hydrogen bonds and van der Waals attractions between silica particles (and alumina sites), which create a transient gel-like structure in the aqueous continuous phase. Consequently, nanoemulsions with elevated Aerosil concentrations exhibit resistance to flow until a critical shear (yield stress) is attained, at which point they undergo facile flow. This rheology is highly beneficial for product stability and application: the formulation remains thick and prevents phase separation during storage, yet it can be spread easily when rubbed.

The oscillatory rheology (Figure 5) further highlights the structural changes induced by Aerosil COK 84. As the silica concentration increases, the nanoemulsion undergoes a transition from a predominantly viscous liquid to a more elastic, solid-like material. It has been demonstrated that with the addition of Aerosil, G′ undergoes a substantial increase that can surpass G″ across the tested frequency range of 0.008–3 Hz. This phenomenon is indicative of the development of a weak gel structure, characterised by predominant elasticity. The network of silica particles endows the system with the capacity to store elastic energy: under small deformations, the structure resists and recoils (high G′), rather than flowing irreversibly. The emergence of a frequency-independent plateau in G′ at low frequencies for the samples formulated with the fumed silica (as illustrated in Figure 5) is indicative of a percolated network or gel within the sample. Consequently, the loss tangent (G″/G′) decreases with silica content, indicating a transition towards solid-like behaviour. These viscoelastic trends are consistent with a particle-bridged droplet network in the emulsions. In this study, it is hypothesised that Aerosil particles in the continuous phase connect via weak bonds and possibly anchor to droplet interfaces, creating an elastic cage that traps the oil droplets. This finding is consistent with reports that the incorporation of colloidal silica into emulsions results in a substantial augmentation of the yield stress and plateau modulus when compared to formulations devoid of silica. It can thus be concluded that Aerosil functions as a structuring agent, thereby converting a purely viscous emulsion into a viscoelastic gel. In contrast, the combination of ZnO and oil alone did not yield the robust gel-like moduli that were observed with silica. It is evident that the supplementation of Aerosil is pivotal in the customisation of the linear viscoelastic response.

The physical stability of the nanoemulsions, quantified by the Turbiscan Stability Index (TSI) in Table 3, improves markedly with increasing Aerosil content. Conversely, a lower TSI is indicative of reduced phase separation or creaming over time. The data demonstrate that samples devoid of Aerosil exhibit elevated TSI values, indicative of diminished stability. Conversely, samples incorporating Aerosil demonstrate reduced TSI values, suggesting enhanced stability. This enhancement can be directly attributed to the rheological and microstructural effects discussed above. In essence, the arrest of the mobility of droplets and solid particles is induced by silica. Droplet creaming and sedimentation are greatly suppressed because the gel character can counteract the gravitational force on the dispersed particles.

The progressive enrichment of the optimised linseed oil/ZnO nanoemulsion with Aerosil COK 84 resulted in a measurable enhancement of its photo-protective performance. As demonstrated in Table 3, the in vitro SPF exhibited an increase with the incorporation of Aerosil, with statistical significance attained at *p* < 0.05. Despite the fact that the absolute gains are modest in comparison to the increase achieved by ZnO itself, they are noteworthy for two reasons. Firstly, the rise occurs without altering either the active ZnO dose or the oil/surfactant ratio, evidencing a genuine synergistic contribution from the fumed silica/alumina particles. Secondly, the incremental SPF is accompanied by markedly lower TSI values and a rheology shift from a nearly Newtonian fluid to a weak-gel network. Collectively, these factors delay the sedimentation of ZnO and immobilise droplets, thereby preserving the optical homogeneity that is essential for reproducible UV screening.

Figure 6 provides a direct microstructural comparison between the silica-free nanoemulsion (Figure 6A) and its counterpart enriched with 1 wt.% Aerosil COK 84 (Figure 6B). In micrograph Figure 6A, the oil droplets are distinctly delineated by a thin, continuous corona of adsorbed ZnO nanoparticles, giving rise to a classic Pickering shell. The individual droplets remain largely isolated, and the inter-droplet spaces appear dark and unfilled, indicating that the particle layer is confined to the oil/water interface. The configuration under discussion stabilises the dispersion against coalescence. However, the absence of an interconnecting particle scaffold leaves significant voids through which droplets may still migrate under gravity or applied shear. The addition of fumed silica/alumina (Figure 6B) transforms this discrete architecture into a densely interconnected network. While ZnO continues to armour each droplet, Aerosil aggregates are now observed to be bridging neighboring interfaces and populating the continuous phase, thereby producing a space-filling, fractal-like skeleton. Such a percolated silica framework immobilises both droplets and ZnO particles, thereby accounting for the pronounced rise in low-frequency elasticity, the order-of-magnitude jump in consistency index and the halving of the Turbiscan Stability Index reported elsewhere in the manuscript.

Figure 6 provides a high-resolution perspective on the hierarchical particle architecture of the optimised nanoemulsion (scale bar = 500 nm). As illustrated in panel A, each linseed oil droplet is coated by a continuous, ≈20 nm corona of ZnO nanoparticles that form a classic Pickering shell, while the inter-droplet space remains essentially void of solids. Panel B, obtained following the incorporation of 1 wt.% Aerosil COK 84, displays a remarkably divergent morphology: conspicuous, ramified aggregates of fumed silica/alumina percolate through the continuous phase, thereby bridging neighboring ZnO-armoured droplets, thereby creating a web-like scaffold. The dual-level structure, composed of interfacial ZnO shells and a bulk Aerosil network, rationalises the sharp rise in low-frequency G′ and the halving of the Turbiscan Stability Index, as discussed previously. This confirms that silica/alumina functions as a load-bearing matrix, immobilising droplets and enhancing multiple scattering.

## 4. Conclusions

The formulation of a linseed oil/ZnO nanoemulsion in conjunction with the biodegradable surfactant Appyclean 6552 provides a fundamental level of photoprotection, which can be significantly enhanced through the incorporation of trace amounts of fumed silica/alumina. In the first instance, the central composite response surface design was utilised to identify a linseed oil/ZnO/Appyclean composition that simultaneously minimised mean droplet size (<300 nm) and maximised baseline in vitro SPF (≈3.5). The addition of Aerosil COK 84 in increasing concentrations, from 0 to 2 wt.%, results in the transformation of the initially Newtonian dispersion into a weak elastic gel. This process concomitantly causes a halving of the Turbiscan Stability Index and a 21% increase in the in vitro SPF, without the necessity of additional ZnO. Field-emission SEM analysis confirms a hierarchical architecture, in which ZnO nanoparticles armour individual droplets while percolated silica aggregates bridge neighboring interfaces, immobilising the dispersed phase and enhancing multiple UV scattering. The dual particle mechanism provides a physicochemical rationale for simultaneously improving stability and photoprotection, offering a transferable, fully mineral route to greener sunscreen formulations based on plant oils.

## Figures and Tables

**Figure 1 materials-18-03030-f001:**
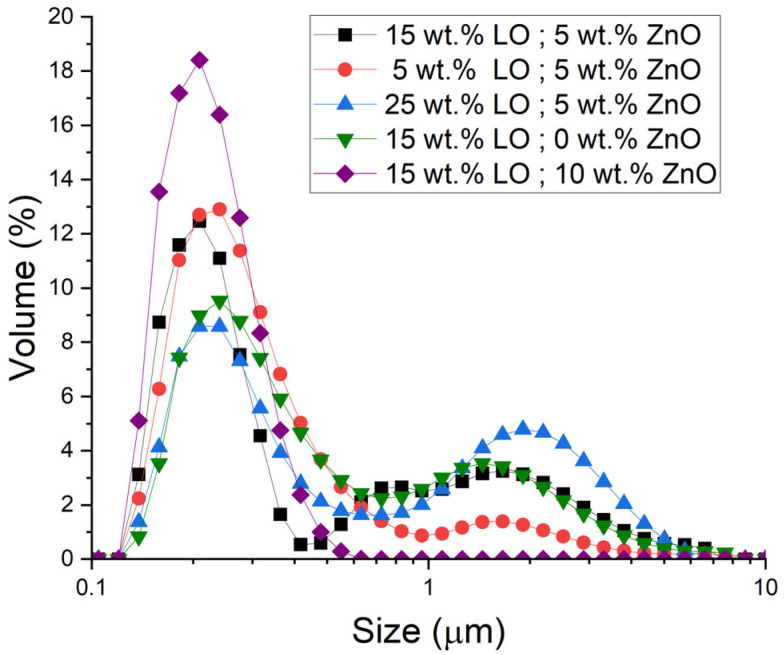
Droplet size distributions of selected nanoemulsions as a function of linseed oil (LO) concentration and zinc oxide (ZnO) concentration.

**Figure 2 materials-18-03030-f002:**
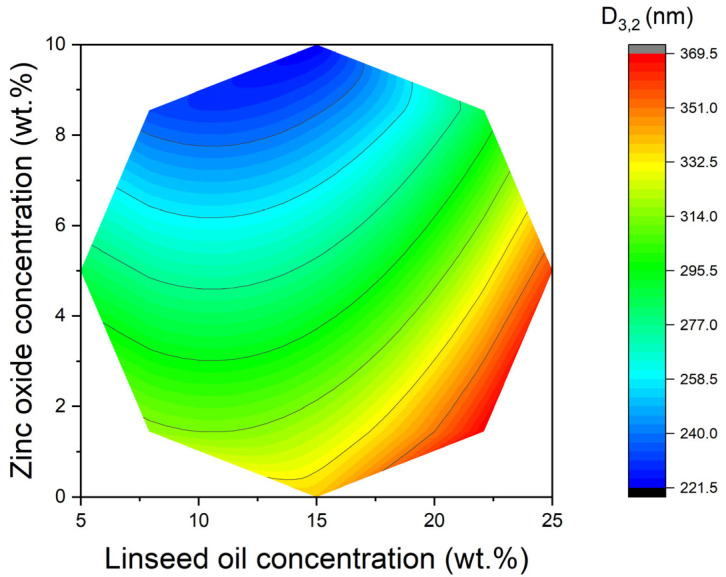
2D representation of the Sauter mean diameter (D_3,2_) with respect to linseed oil (LO) concentration and zinc oxide concentration (ZnO) according to the quadratic model obtained by the response surface methodology.

**Figure 3 materials-18-03030-f003:**
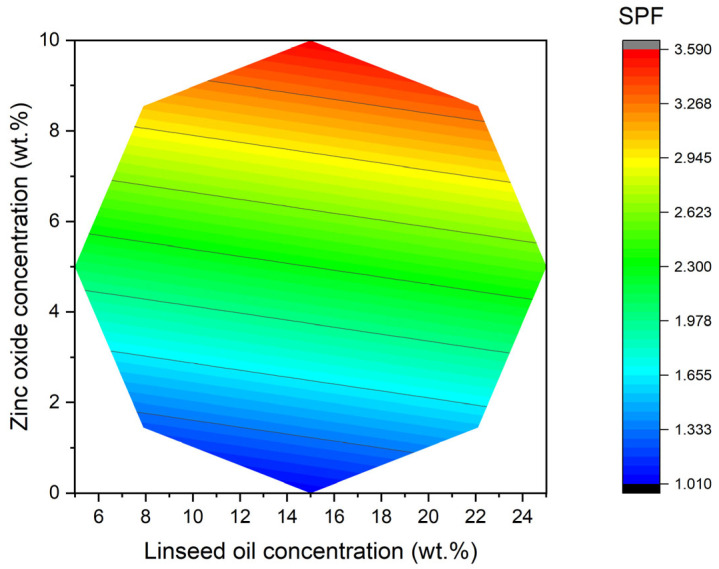
2D representation of the sun protection factor (SPF) with respect to linseed oil (LO) concentration and zinc oxide concentration (ZnO) according to the quadratic model obtained by the response surface methodology.

**Figure 4 materials-18-03030-f004:**
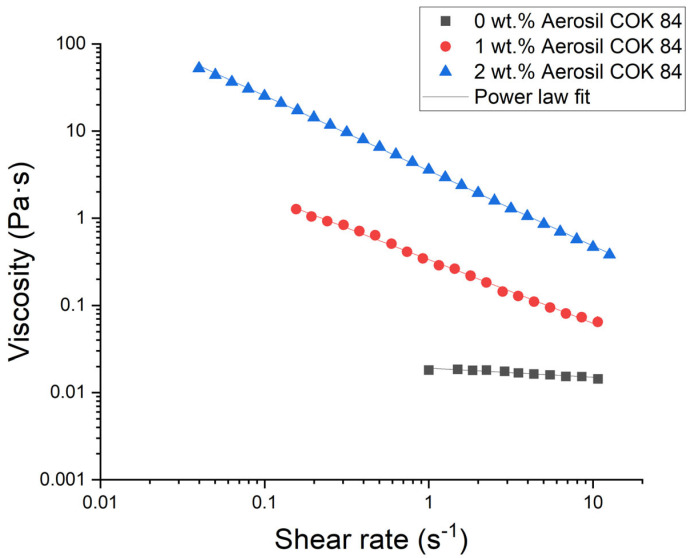
Flow curves of nanoemulsions as a function of Aerosil COK 84 concentration.

**Figure 5 materials-18-03030-f005:**
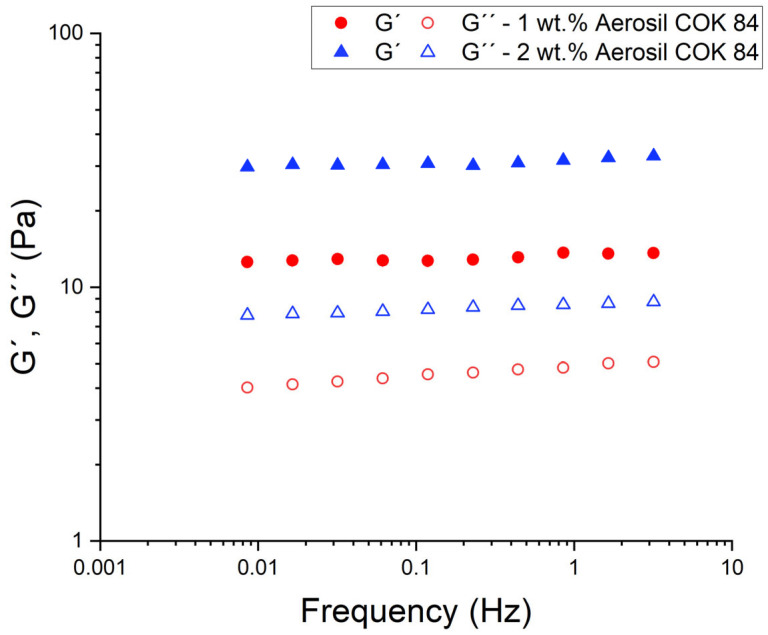
Mechanical spectra as a function of Aerosil COK 84 concentration.

**Figure 6 materials-18-03030-f006:**
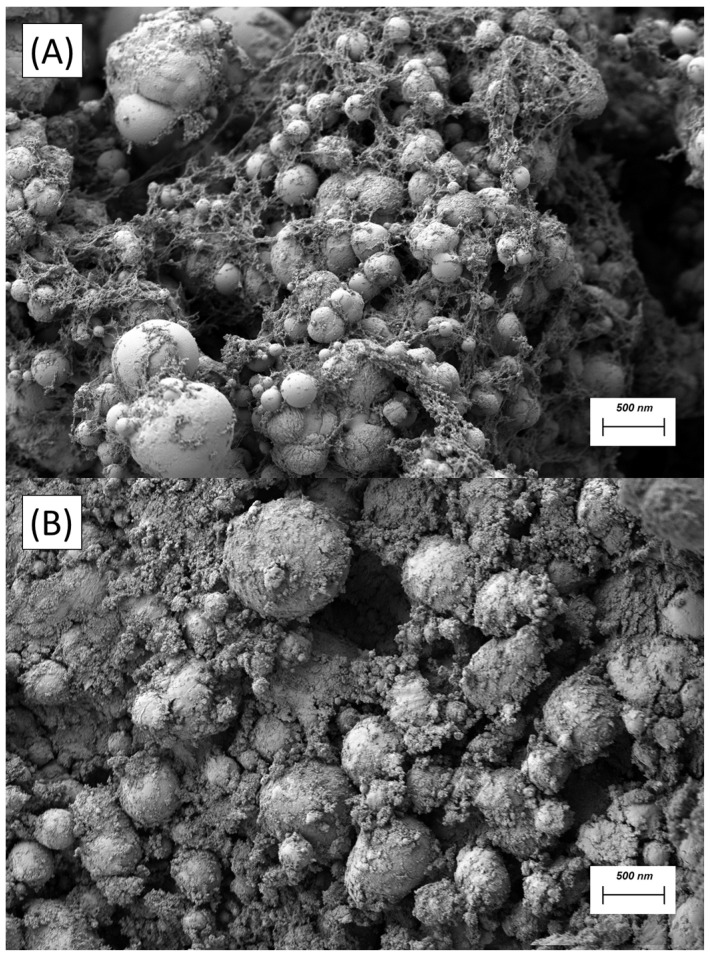
FESEM micrograph of nanoemulsions formulated with 15 wt.% linseed oil, 10 wt.% ZnO and (**A**) 0 wt.% Aerosil COK 85 and (**B**) 1 wt.% Aerosil COK 84.

**Table 1 materials-18-03030-t001:** Experimental design used with indication of coded and actual values in weight percentage (wt.%) for the concentration of linseed oil (LO) and zinc oxide (ZnO).

Sample	Linseed Oil Concentration (LO)	Zinc Oxide Concentration (ZnO)	Linseed Oil Concentration (wt.%)	Zinc Oxide Concentration (wt.%)
1	−1	−1	7.9	1.45
2	1	−1	22.1	1.45
3	−1	1	7.9	8.55
4	1	1	22.1	8.55
5	−1.41	0	5	5
6	1.41	0	25	5
7	0	−1.41	15	0
8	0	1.41	15	10
9	0	0	15	5
10	0	0	15	5

**Table 2 materials-18-03030-t002:** Sauter mean diameter (D_3,2_) and sun protection factor (SPF) of the nanoemulsions as a function of the concentration of linseed oil and zinc oxide.

Sample	Linseed Oil Concentration (wt.%)	Zinc Oxide Concentration (wt.%)	D_3,2_ (nm)	SPF
1	7.9	1.45	325	2.28
2	22.1	1.45	355	2.45
3	7.9	8.55	259	3.05
4	22.1	8.55	285	3.36
5	5	5	261	2.21
6	25	5	370	2.60
7	15	0	339	0.39
8	15	10	200	3.51
9	15	5	286	2.34
10	15	5	275	2.37

**Table 3 materials-18-03030-t003:** Consistency index (k), flow index (n), Turbiscan Stability Index (TSI) and sun protection factor (SPF) of the nanoemulsions as a function of the Aerosil COK 84 concentration.

Aerosil COK 84 (wt.%)	k (Pa·s^n^)	n	TSI	SPF
0	0.0191 ± 0.0003 ^a^	0.90 ± 0.01 ^a^	4.2 ± 0.4 ^a^	3.51 ± 0.08 ^a^
1	0.3343 ± 0.0079 ^b^	0.27 ± 0.01 ^b^	1.3 ± 0.2 ^b^	4.00 ± 0.15 ^b^
2	3.4978 ± 0.1588 ^c^	0.14 ± 0.01 ^c^	1.2 ± 0.2 ^b^	4.26 ± 0.12 ^b^

^a,b,c^ Standard deviation with statistically significant (*p* < 0.05) difference according to Tukey’s test.

## Data Availability

The raw data supporting the conclusions of this article will be made available by the authors on request.

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
