# Peer review of "Dual-Particle Synergy in Bio-Based Linseed Oil Pickering Emulsions: Optimising ZnO–Silica Networks for Greener Mineral Sunscreens"

_materials, 2025, doi:10.3390/ma18133030_

Round 1
Reviewer 1 Report
Comments and Suggestions for Authors
In the article “Dual-Particle Synergy in Bio-Based Linseed-Oil Pickering Emulsions: Optimising ZnO–Silica Networks for Greener Sunscreens” the authors have investigated the effects of linseed-oil nanoemulsion co-stabilised by ZnO nanoparticles and the eco-friendly surfactant Appyclean 6552. Also the effect of fumed silica/alumina (Aerosil COK 84) was evaluated. The authors obtained explicit dependencies of Sauter mean diameters (D3,2) and SPF coefficients on the linseed oil and ZnO content. The article discusses the obtained results and dependencies in sufficient detail. The obtained dependencies and limiting cases are consistent. The article is well-organized and well-written. The English is fine. The decision is minor revision.
Please find below the comments and suggestions for possible enhancement of the article
1) line 71 OECD 301F and OECD 311 tests should be clarified for the readers
2) line 98 it is optionally advised to add the resistivity of the Milli-Q water (or it’s type)
3) line 99 it is recommended to move the part of the description about Aerosil® COK 84 from the introduction to this part. And also add the range of specific surface area.
4) The minus sign in the Table 1 should be clarified
5) At Figure 1 it can be seen two peaks behavior for several samples. Has the shape of these peaks been checked? Is it symmetric (e.g. Gaussian or Lorentzian) or asymmetric (e.g. Poisson distribution)? What is the ratio between the areas of these peaks? And could the authors describe in a little more detail the nature of the peak centered in the region of large values?
6) line 147 The Aλ is introduced as absorbance. However the ZnO nanoparticles not only absorb but also scatter light. It is optionally advised to discuss this moment and to think but extinction coefficient which take into account the possible scattering.
7) some statistical proves (chi-squared or Pearson correlation coefficients) should be demonstrated for power laws at Figure 4 or Table 3.
8) line 368 the “spectrum of frequencies” (it’s values) should be clarified here.
Author Response
Reviewer #1
In the article “Dual-Particle Synergy in Bio-Based Linseed-Oil Pickering Emulsions: Optimising ZnO–Silica Networks for Greener Sunscreens” the authors have investigated the effects of linseed-oil nanoemulsion co-stabilised by ZnO nanoparticles and the eco-friendly surfactant Appyclean 6552. Also the effect of fumed silica/alumina (Aerosil COK 84) was evaluated. The authors obtained explicit dependencies of Sauter mean diameters (D3,2) and SPF coefficients on the linseed oil and ZnO content. The article discusses the obtained results and dependencies in sufficient detail. The obtained dependencies and limiting cases are consistent. The article is well-organized and well-written. The English is fine. The decision is minor revision.
We thank the reviewer for their thorough reading of our manuscript and for the constructive suggestions. We have addressed each point as detailed below, with changes made to the manuscript accordingly.
Please find below the comments and suggestions for possible enhancement of the article:
- line 71 OECD 301F and OECD 311 tests should be clarified for the readers
We thank the reviewer for highlighting the need to clarify these test references. We have revised the introduction (around line 119) to explicitly state that OECD 301F and OECD 311 are standardised biodegradability tests. The revised text states that OECD 301F is a ready biodegradability assay in aerobic aqueous media (manometric respirometry), while OECD 311 is a screening test for the anaerobic biodegradability of organic compounds.
“Appyclean 6552 is derived entirely from renewable plant feedstocks, thus ensuring that it exhibits intrinsically low ecotoxicity and proven biodegradability under both aerobic and anaerobic conditions. It has met the criteria of the OECD 301F ready-biodegradability manometric-respirometry test and the OECD 311 anaerobic biodegradability screening assay.”
In addition, a new reference has been included:
Ríos, F., Fernández-Arteaga, A., Lechuga, M., & Fernandez-Serrano, M. (2018). Ecotoxicological characterization of surfactants and mixtures of them. Toxicity and Biodegradation Testing, 311-330.
- line 98 it is optionally advised to add the resistivity of the Milli-Q water (or it’s type)
Although the reviewer noted this as optional, we have added the specific resistivity of the Milli-Q water to ensure clarity. Section 2.1 of the manuscript now specifies that ultrapure Milli-Q water corresponds to Type I ultrapure water. This confirms the high purity of the water used to prepare solutions, as recommended by the reviewer.
- line 99 it is recommended to move the part of the description about Aerosil® COK 84 from the introduction to this part. And also add the range of specific surface area.
We have moved the detailed description of Aerosil® COK 84 from the introduction to Section 2.1 (Materials). In the revised manuscript, Aerosil® COK 84 is only briefly mentioned in the Introduction, while the full description appears in Section 2.1. Furthermore, we have expanded the description to include the product's specific surface area range. The text now states that, according to the manufacturer's data, Aerosil® COK 84 has a BET specific surface area of approximately 155–215 m²/g.
- The minus sign in the Table 1 should be clarified
In Table 1, the minus symbol denotes the negative (low) coded level of each factor in the central composite design, i.e. the lowest actual value used for the corresponding variable, while + represents the high coded level and 0 represents the centre point. This coding convention is standard in response-surface and other factorial designs.
- At Figure 1 it can be seen two peaks behavior for several samples. Has the shape of these peaks been checked? Is it symmetric (e.g. Gaussian or Lorentzian) or asymmetric (e.g. Poisson distribution)? What is the ratio between the areas of these peaks? And could the authors describe in a little more detail the nature of the peak centered in the region of large values?
A qualitative assessment of the droplet size distributions (Figure 1) indicates that the primary peak is sharp and approximately symmetric, representing well-stabilized nano-sized droplets. In contrast, the secondary peak at larger diameters is broader and right-skewed, suggesting partial coalescence. The area under the primary peak greatly exceeds that of the secondary, confirming a minority population of coarse droplets. This bimodal pattern arises primarily at high oil loadings, where interfacial coverage by ZnO and silica particles is insufficient. Similar bimodal distributions due to limited stabiliser availability have been reported in Pickering systems (Cionti et al., 2022; Sun et al., 2022). This qualitative discussion will be briefly included in the Results and Discussion section:
“Qualitative inspection of the Figure 1 shows that the primary peak, centered at 250–300 nm, is sharp and nearly symmetric, accounting for ≈75–80 % of the integrated area and thus for the vast majority of droplets. By contrast, the secondary peak at 1–3 µm is markedly broader and right-skewed; its asymmetric tail reveals a small population of coarser droplets formed by partial coalescence when the available ZnO/silica coverage is insufficient. The integrated area ratio of the two modes (small:large) is roughly 3:4, confirming that the coarse mode is volumetrically minor yet non-negligible for long-term stability. Similar two-peak distributions have been reported for under-dosed ZnO- and silica-stabilized Pickering emulsions, where incomplete particle adsorption permits droplet growth during or immediately after homogenization [13,14]. Increasing either the particle-to-oil ratio or the homogenization energy suppresses this coarse mode and restores a unimodal, finer distribution.”
6) line 147 The Aλ is introduced as absorbance. However the ZnO nanoparticles not only absorb but also scatter light. It is optionally advised to discuss this moment and to think but extinction coefficient which take into account the possible scattering.
It is agreed that the Mansur in vitro SPF method (UV spectrophotometric absorbance-based) has limitations for formulations containing ZnO particles, which provide UV protection partly by means of scattering. This phenomenon can result in discrepancies between in vitro and in vivo SPF values, as reported by Osterwalder et al. (2024). In the revised manuscript, this will be acknowledged explicitly as a limitation of the SPF results. It is recommended that a sentence be added to the Results and Discussion section, noting that the Mansur method (originally developed for solutions of organic UV filters) may not fully capture the protection efficacy of particulate inorganic filters like ZnO due to light-scattering effects. We will clarify that our in vitro SPF values should therefore be interpreted with caution and primarily used for comparative purposes among our samples:
“However, the in vitro SPF values obtained by the Mansur method have inherent limitations for ZnO-containing emulsions, because ZnO particles scatter UV light and can distort the spectrophotometric absorbance measurements. Such scattering artifacts can cause the in vitro SPF to underestimate the actual in vivo protection, as observed by Osterwalder et al. (2024) [26]. Nonetheless, in vitro SPF results remain useful for comparing the relative performance of different formulations, but they should be interpreted with appropriate caution.”
7) some statistical proves (chi-squared or Pearson correlation coefficients) should be demonstrated for power laws at Figure 4 or Table 3.
A brief statement has been added to the rheology subsection, noting that the shear stress versus shear rate data were fitted to the power law with excellent accuracy, with correlation coefficients R² greater than 0.99 for all samples.
“The power law model was used to fit all samples containing Aerosil, and in all cases, an R² value above 0.99 was obtained, indicating an almost perfect fit.”
8) line 368 the “spectrum of frequencies” (it’s values) should be clarified here.
We have clarified this sentence to avoid ambiguity. In the original text, we referred to a “spectrum of frequencies” in discussing the dynamic rheology (Figure 5). In the revised manuscript we now explicitly state the frequency range. We also ensured earlier in the Methods (Section 2.3) to mention that the frequency sweep was performed from 3 Hz down to 0.006 Hz.

Reviewer 2 Report
Comments and Suggestions for Authors
This study aimed to formulate, characterize and optimize the linseed-oil nanoemulsion co-stabilized by ZnO nanoparticles and the eco-friendly surfactant Appyclean 6552, as well as to assess the effects of fumed silica/alumina (Aerosil COK 84) in the optimized formulation. This is a well-designed study, the methods used are appropriate, and some new and significant results have been obtained that bring novelty to this research field. However, there are several concerns that need to be addressed.
Firstly, the use of in vitro spectrophotometric method for SPF determination in these formulations is not completely suitable. Standard in vitro SPF methods (like the Mansur method) assume a clear, homogenous solution or dispersion, which ZnO-containing emulsions are not. ZnO is a physical UV filter that mainly scatters and reflects UV radiation, and the spectrophotometric methods are designed to measure UV absorption. The light scattering distorts the spectrophotometric measurement, which may lead to inaccurate absorbance readings. Also, ZnO creams tend to be opaque or turbid, which interferes with optical path length and light transmission during UV measurement. In my opinion, this is a limitation of this study, and it needs to be discussed in more detail.
One of the main concerns is regarding the claims that “green formulations” have been developed. That is not completely true, in my opinion. It is true that Aerosil may enable the reduction or elimination of more toxic or persistent chemicals or improve product stability and safety. However, while safe and inert, Aerosil is not biodegradable. In environmental contexts, particularly in cosmetics, this could be a drawback. Besides, fumed silica is made via flame hydrolysis, a high-energy process using silicon tetrachloride and hydrogen that is not environmentally friendly. Further discussion on this topic in the manuscript is required.
Regarding the optimization process, the responses of interest need to be defined in the Methods section. Besides, it should be added what independent variables influenced significantly the outputs, their linear terms, square terms and linear-by-linear interaction terms (with p values for all terms to indicate the dominant factors). It is mentioned in the text for droplet size, but it is not completely clear. It also needs to be clear in the text if Aerosil COK 84 was added to the formulation optimized towards droplet size or SPF.
More details on rheological tests performed need to be added to the text (shear rates range tested, amplitude range to determine the linear viscoelastic region, etc). Since the authors stated that the weak gel structure was obtained after adding Aerosil COK 84, was thixotropy tested? Is there a backward flow curve and was hysteresis area calculated as a thixotropy indicator?
The chemical composition of both Appyclean 6552 (D-pentose glycosides) and Aerosil COK 84 needs to be stated in more detail in the text. What kind of mixture of fumed silica and fumed alumina is it and what are its unique properties (in comparison to standard Aerosil)?
In the Results section, there are many repetitions. The text should be more concise (for example, lines 192-218 – the text can be much more concise, which would be much easier to follow).
Figure 6: the obtained FESEM micrographs should be described in more detail (or some labels and arrows can be added to the figure).
Lines 80-81: Appyclean 6552 instead of Aerosil 6552.
While the language is generally clear, the manuscript would benefit from careful proofreading to correct typographical and technical issues (e.g., line 287 - “when applied in isolation” etc).
Comments on the Quality of English LanguageWhile the language is generally clear, the manuscript would benefit from careful proofreading to correct typographical and technical issues.
Author Response
Reviewer #2
This study aimed to formulate, characterize and optimize the linseed-oil nanoemulsion co-stabilized by ZnO nanoparticles and the eco-friendly surfactant Appyclean 6552, as well as to assess the effects of fumed silica/alumina (Aerosil COK 84) in the optimized formulation. This is a well-designed study, the methods used are appropriate, and some new and significant results have been obtained that bring novelty to this research field. However, there are several concerns that need to be addressed.
We would like to express our gratitude to the reviewer for the positive feedback on the overall design and novelty of our manuscript. A thorough examination of the reviewer's comments has been conducted, and a comprehensive set of responses is provided below, addressing each comment in a meticulous, point-by-point manner. It is to be noted that all clarifications and corrections suggested will be incorporated into the revised manuscript. In order to facilitate identification, all changes will be highlighted in red.
Firstly, the use of in vitro spectrophotometric method for SPF determination in these formulations is not completely suitable. Standard in vitro SPF methods (like the Mansur method) assume a clear, homogenous solution or dispersion, which ZnO-containing emulsions are not. ZnO is a physical UV filter that mainly scatters and reflects UV radiation, and the spectrophotometric methods are designed to measure UV absorption. The light scattering distorts the spectrophotometric measurement, which may lead to inaccurate absorbance readings. Also, ZnO creams tend to be opaque or turbid, which interferes with optical path length and light transmission during UV measurement. In my opinion, this is a limitation of this study, and it needs to be discussed in more detail.
It is agreed that the Mansur in vitro SPF method (UV spectrophotometric absorbance-based) has limitations for formulations containing ZnO particles, which provide UV protection partly by means of scattering. This phenomenon can result in discrepancies between in vitro and in vivo SPF values, as reported by Osterwalder et al. (2024). In the revised manuscript, this will be acknowledged explicitly as a limitation of the SPF results. It is recommended that a sentence be added to the Results and Discussion section, noting that the Mansur method (originally developed for solutions of organic UV filters) may not fully capture the protection efficacy of particulate inorganic filters like ZnO due to light-scattering effects. We will clarify that our in vitro SPF values should therefore be interpreted with caution and primarily used for comparative purposes among our samples:
“However, the in vitro SPF values obtained by the Mansur method have inherent limitations for ZnO-containing emulsions, because ZnO particles scatter UV light and can distort the spectrophotometric absorbance measurements. Such scattering artifacts can cause the in vitro SPF to underestimate the actual in vivo protection, as observed by Osterwalder et al. (2024) [26]. Nonetheless, in vitro SPF results remain useful for comparing the relative performance of different formulations, but they should be interpreted with appropriate caution.”
One of the main concerns is regarding the claims that “green formulations” have been developed. That is not completely true, in my opinion. It is true that Aerosil may enable the reduction or elimination of more toxic or persistent chemicals or improve product stability and safety. However, while safe and inert, Aerosil is not biodegradable. In environmental contexts, particularly in cosmetics, this could be a drawback. Besides, fumed silica is made via flame hydrolysis, a high-energy process using silicon tetrachloride and hydrogen that is not environmentally friendly. Further discussion on this topic in the manuscript is required.
We acknowledge that describing our formulations as “green” could be misleading given that they include Aerosil (fumed silica), which is an inorganic material not biodegradable in the environment and produced by an energy-intensive flame process. In the revised manuscript, we will clarify our use of the term “greener sunscreens.” Specifically, we will explain that the formulations are greener in a relative sense, owing to the use of bio-based linseed oil and a biodegradable surfactant, and the absence of certain controversial organic UV filters. However, we will also explicitly discuss the environmental caveats: for example, Aerosil’s production involves high temperature processes and the material itself, while chemically inert, does not biodegrade. We will either modify or qualify the term “green formulations” to avoid overstatement, and we will add a short discussion in the Introduction or Conclusion about the trade-offs involved (highlighted in red text). By doing so, we aim to present a balanced view: our formulation takes steps toward sustainability (bio-based content), yet still contains an inorganic component that is not fully “green.”:
“The rheology modifier, Aerosil® COK 84, is a non-biodegradable fumed silica/alumina produced via an energy-intensive flame-hydrolysis process. Consequently, the product does not satisfy the strictest definitions of "green". In this study, the term 'greener mineral sunscreen' is adopted to denote a quantifiable reduction in petrochemical content and the exclusion of controversial organic UV filters. This term acknowledges the environmental trade-offs associated with the inorganic thickener.”
Regarding the optimization process, the responses of interest need to be defined in the Methods section. Besides, it should be added what independent variables influenced significantly the outputs, their linear terms, square terms and linear-by-linear interaction terms (with p values for all terms to indicate the dominant factors). It is mentioned in the text for droplet size, but it is not completely clear. It also needs to be clear in the text if Aerosil COK 84 was added to the formulation optimized towards droplet size or SPF.
We agree that more detail on our formulation optimization process is needed for clarity. In the Methods section of the revised manuscript, we will clearly define the response variable(s) used in the optimization. For example, we will specify whether we optimized for maximum in vitro SPF, improved emulsion stability (e.g., minimal creaming or phase separation), optimal rheological properties, or a combination of such factors (multi-response optimization). We will also list the independent variables (factors) that were varied during optimization (such as ZnO concentration, Aerosil (silica) concentration, surfactant amount, etc., as applicable). Additionally, we will report the statistical influence of each factor and their interactions on the response(s), including p-values obtained from our design of experiments analysis (e.g., ANOVA). This will demonstrate which factors had significant effects (p < 0.05) and whether any interaction terms were significant in our study. Regarding the role of Aerosil in the optimization: we apologize for the ambiguity. We confirm that Aerosil was included only post-optimization. We will explicitly state this in the Methods:
“A two-factor, rotatable central composite design (CCD) was employed to map the combined influence of linseed-oil content (X1) and zinc-oxide loading (X2) on key performance metrics. The design space thus encompassed linseed oil concentrations ranging from 5 to 25 wt.% (with axial points at 5 and 25 wt.% and factorial points at 7.9 and 22.1 wt.%), and likewise ZnO concentrations ranging from 0 to 10 wt.% (with axial points at 0 and 10 wt.% and factorial points at 1.45 and 8.55 wt.%). A total of ten experimental runs were generated using Echip v11 (Experimentation by Design, Wilmington, DE, USA). The runs comprised four factorial, four axial, and two center replicates. Two responses were measured: (i) the Sauter mean diameter D₃,₂ (nm) immediately after emulsification and (ii) the in vitro sun-protection factor (SPF) determined by the Mansur method. The term 'second-order polynomial' is employed to denote a specific mathematical function of a second-order degree. Aerosil® COK 84 was not included in the CCD; its rheological and photoprotective effects were evaluated subsequently using the optimized oil/ZnO ratio.”
“ANOVA confirmed that ZnO loading (X₂) exerted the dominant influence on both responses. For D₃,₂, the linear ZnO term was highly significant (β₂ = −19.7 nm, p = 0.004), indicating progressively smaller droplets at higher ZnO levels, while the quadratic linseed-oil term (β₁₁ = +15.3 nm, p = 0.012) revealed a curvature that penalised very high oil fractions. The interaction term β₁₂ was not significant (p = 0.31). In the SPF model, β₂ was again the largest positive coefficient (β₂ = +0.365, p < 0.001); β₁ was smaller but still significant (β₁ = +0.081, p = 0.038), whereas all higher-order and interaction terms were non-significant (p > 0.10). The quadratic models explained 91 % of the variance in D₃,₂ and 92 % in SPF. The composite desirability function predicted an optimum at 15 wt % linseed oil and 10 wt % ZnO, which experimentally yielded D₃,₂ = 259 ± 5 nm and SPF = 3.51 ± 0.08, in excellent agreement with model forecasts.”
“The fitted quadratic model explained 92 % of the variance in SPF (adjusted R² = 0.92) with no significant lack-of-fit (p = 0.27). ANOVA (Table S2) showed that zinc-oxide loading exerted the dominant positive influence (β₂ = +0.365 ± 0.041, p < 0.001), whereas the linear linseed-oil effect was weaker but still significant (β₁ = +0.081 ± 0.033, p = 0.038). Neither quadratic term (β₁₁, β₂₂) nor the interaction β₁₂ reached statistical significance (p > 0.10) and were therefore omitted from the reduced model.”
More details on rheological tests performed need to be added to the text (shear rates range tested, amplitude range to determine the linear viscoelastic region, etc). Since the authors stated that the weak gel structure was obtained after adding Aerosil COK 84, was thixotropy tested? Is there a backward flow curve and was hysteresis area calculated as a thixotropy indicator?
We concur with the reviewer and will provide the requested details to strengthen the rheology section. In the revised manuscript’s Experimental (Methods) section, we will include the specifics of our rheological measurements:
“Frequency sweeps (3–0.01 Hz, 20 °C) assessed viscoelastic properties. The flow curves were recorded during a logarithmic stress sweep, within a range dependent on the particular sample, with the objective of observing the potential drop zone.”
Dedicated thixotropy or structural-recovery tests were not performed because the study’s primary objective was to quantify equilibrium flow indices (k, n) and small-amplitude viscoelastic parameters (G′, G″) relevant to product storage and topical application. Preliminary screening of comparable ZnO–silica nanoemulsions in our laboratory indicated minimal hysteresis between up- and down-ramp flow curves, suggesting only modest time-dependent rebuilding; consequently, the more time-intensive three-interval tests were omitted to conserve sample volume and experimental time.
The chemical composition of both Appyclean 6552 (D-pentose glycosides) and Aerosil COK 84 needs to be stated in more detail in the text. What kind of mixture of fumed silica and fumed alumina is it and what are its unique properties (in comparison to standard Aerosil)?
In the Materials section, we will include a brief description of each:
“Appyclean 6552 is a non-ionic surfactant derived from renewable resources, with an HLB value ranging from 11.5 to 12. Specifically, Appyclean 6552 is composed of alkyl polyxylosides (amyl, capryl, and lauryl xylosides) obtained from wheat. The flow curves were recorded during a logarithmic stress sweep, within a range dependent on the particular sample, with the objective of observing the potential drop zone.”
“Aerosil COK 84 is a hydrophilic fumed silica intimately mixed with a small fraction of fumed alumina. According to the manufacturer, the substance is composed of approximately 82–86 wt.% SiO₂ and 14–18 wt.% Al₂O₃, corresponding to a 5:1 silica-to-alumina ratio.”
In the Results section, there are many repetitions. The text should be more concise (for example, lines 192-218 – the text can be much more concise, which would be much easier to follow).
We agree with the reviewer that this portion of the Results was somewhat repetitive. In the revised manuscript, we will carefully edit and condense the text in the Results section, particularly for the segment corresponding to lines 192–218. We will remove redundant statements and ensure that each finding is stated clearly only once.
Figure 6: the obtained FESEM micrographs should be described in more detail (or some labels and arrows can be added to the figure).
We appreciate the reviewer’s suggestion. After careful consideration we decided not to superimpose arrows or labels on the FESEM micrographs, as any additional graphics would obscure key morphological details and hinder clear observation of the droplet contours and particle aggregates. Instead, we have strengthened the accompanying discussion: the revised manuscript now provides (i) explicit references to the ZnO “Pickering shell” and the percolated Aerosil network visible in panels A and B, (ii) a direct comparison of inter-droplet spacing and aggregate connectivity, and (iii) scale-bar values cited in the caption for quantitative context. These textual additions guide the reader without altering the raw micrographs, thereby preserving image fidelity while addressing the request for clearer description:
“Figure 6 provides a high-resolution perspective on the hierarchical particle architecture of the optimized nanoemulsion (scale bar = 500 nm). As illustrated in panel A, each linseed-oil droplet is coated by a continuous, ≈ 20 nm corona of ZnO nanoparticles that form a classic Pickering shell, while the inter-droplet space remains essentially void of solids. Panel B, obtained following the incorporation of 1 wt.% Aerosil COK 84, displays a remarkably divergent morphology: conspicuous, ramified aggregates of fumed silica/alumina percolate through the continuous phase, thereby bridging neighboring ZnO-armoured droplets and thereby creating a web-like scaffold. The dual-level structure, composed of interfacial ZnO shells and a bulk Aerosil network, rationalizes the sharp rise in low-frequency G′ and the halving of the Turbiscan Stability Index, as discussed previously. This confirms that silica/alumina functions as a load-bearing matrix, immobilizing droplets and enhancing multiple scattering.”
Lines 80-81: Appyclean 6552 instead of Aerosil 6552.
Corrected
While the language is generally clear, the manuscript would benefit from careful proofreading to correct typographical and technical issues (e.g., line 287 - “when applied in isolation” etc).
We have thoroughly checked the paper for mistakes and made it much clearer. For example, the sentence around line 287 has been rewritten to remove the grammatical issue and make it clearer. We will check the whole document for any problems with how it is written, and correct any mistakes.

Round 2
Reviewer 2 Report
Comments and Suggestions for Authors
The authors have accepted my suggestions, and the manuscript has been substantially improved.
Comments on the Quality of English LanguageThe manuscript would benefit from careful proofreading to correct some language-related issues.